# Chlorogenic Acid of *Cirsium japonicum* Resists Oxidative Stress Caused by Aging and Prolongs Healthspan via SKN-1/Nrf2 and DAF-16/FOXO in *Caenorhabditis elegans*

**DOI:** 10.3390/metabo13020224

**Published:** 2023-02-03

**Authors:** Myogyeong Cho, Yebin Kim, Sohyeon You, Dae Youn Hwang, Miran Jang

**Affiliations:** 1Department of Food Technology and Nutrition, Inje University, Gimhae 50834, Republic of Korea; 2Bio-Health Convergence, Duksung Women’s University, Seoul 01369, Republic of Korea; 3Department of Biomaterials Science (BK21 FOUR Program), Life and Industry Convergence Research Institute, College of Natural Resources & Life Science, Pusan National University, Miryang 50463, Republic of Korea

**Keywords:** thistle (*Cirsium japonicum*: CJ), chlorogenic acid (CA), reactive oxygen species (ROS), skn-1/Nrf2, daf-16/FOXO, healthspan, *Caenorhabditis elegans*

## Abstract

To evaluate the value of *Cirsium japonicum* (CJ; thistle) as a material for functional foods, we studied the functional composition of cultivated CJ and the in vitro and in vivo antioxidant activity of the functional substance. The detected phenolics in farmed CJ were chlorogenic acid (CA), linarin (LIN), and pectolinarin (PLIN) by HPLC analysis. As a result of the antioxidant activity of CJ and its phenolics by DPPH and ABTS method, CA had shown the greatest antioxidant activity. We employed *Caenorhabditis elegans* to validate that in vitro effects of CA are shown in vivo. CA delayed reduction in pumping rate and progeny production during aging of *C. elegans*. Under both normal and oxidative stress conditions, CA reduced the production of reactive oxygen species (ROS) in worms and increased their lifespan. In particular, CA showed the reducing effect of ROS accumulation due to aging in aged worms (8 days old). To gain insight into the mechanism, we used skn-1/Nrf2 and daf-16/FOXO transformed worms. The CA effects (on catalase activity and lifespan extension) in the wild-type (WT) decreased in skn-1 and daf-16 mutants. In particular, CA strongly relied on daf-16 under mild oxidative condition and skn-1 under overall (from mild to strong) oxidative stress to reduce ROS and extend healthspan. Thus, we conclude that CA, a key bioactive phenolic of CJ, reduces ROS production and ultimately extends healthspan, and this effect is the result of actions of daf-16 or skn-1 at different stages depending on the degree of oxidation or aging. Our results suggest that CJ containing CA can be used as an antiaging material due to its antioxidant properties.

## 1. Introduction

Aging is caused by multifactor stress and is a phenomenon that gradually decreases resistance to disease by degenerating human function [1]. Reactive oxygen species (ROS) overproduction causes damage to proteins, lipids, and DNA, and thus can induce physical, chemical, or functional changes [2,3]. ROS imbalance is regarded as an aging factor and one that could reduce lifespan and adversely affect health [4]. ROS is no longer a simple metabolic byproduct, but a signaling molecule that regulates physiological activities, including molecular pathways, gene expression, cell proliferation, or apoptosis [5,6,7]. According to reports by Choi et al. (2012) and Kelsey et al. (2010), there are positive correlations between the consumption of foods rich in phenolics and reducing ROS, which can protect cells from oxidative damage [2,8].

Human longevity has dramatically increased during the last century, and much effort has been invested in the development of antiaging food supplements and medicines that increase lifespan. Therefore, healthspan has also recently attracted research interest [9]. We used *Caenorhabditis elegans* to evaluate CA effects on in vivo antioxidant activity and longevity. Nematodes are an important model for the study of human diseases and longevity because they have many similar biological characteristics to humans, such as muscles, nerves, and intestines [10]. In particular, nematodes are suitable for research on disease-related lifespan because their entire life is short as 20 days [11].

There are several genetic pathways that contribute to prolonging lifespan, among which skn-1/Nrf2 and daf-16/FOXO are noticed in relation to stress tolerance [12]. These two transcription factors regulate the expression of genetic factors such as *sod*, *cat*, and *gpx*, inducing antioxidant enzyme activity and ultimately resisting oxidative stress [12]. These two transcription factors cooperate with each other or act independently when an antioxidant system needs to be operated, to resist oxidative stress and extend lifespan [13].

*Cirsium japonicum* (CJ, a thistle native to Japan and Korea) belongs to the family Compositae and is well distributed in the wild. The plant is characterized by spiny leaves and purple flowers. CJ is used to treat stomachaches, wounds, boils, and piles, according to the USDA-NRCS Plants Database (2010), and is consumed in Asia to promote metabolic health and maintain homeostasis [14,15]. In particular, in Korea and Japan, CJ is used to make teas and wines and to produce extract tablets. It has been reported that CJ contains phenolic compounds such as chlorogenic acid (CA), linarin (LIN), pectolinarin (PLIN), cirsmaritin, cirsmarin, acacetin, luteolin, apigenin, hispidulin, and silymarin [16,17,18,19,20]. According to various studies, CJ phenolics have antioxidative, anti-inflammatory, antiobesity, antidiabetic, and antimenopause effects [16,17,21,22,23,24,25].

Chlorogenic acid (CA) is widely distributed in plants, fruits, and vegetables [26], and has been reported to exhibit various health benefits, having antioxidant, anti-inflammatory, antimicrobial, anticardiovascular, and anticancer properties [17,27,28,29,30]. In particular, CA is involved in a comprehensive antioxidation mechanism, directly scavenging free radicals by its polyhydroxy structure, and activating the oxidative signaling pathway by regulating gene expression and endogenous oxidase systems [30]. However, the ROS accumulation caused by aging itself and the intervention of CA during the progression of aging have not been reported. Therefore, we focused on the prevention effect of CA against aging-induced oxidative stress.

In this study, we monitored the phenolic content of CJ using high-performance liquid chromatography–diode-array detector (HPLC-DAD) analysis for three years, and investigated the antioxidant properties of CJ extract and the phenolics it contains. We determined that the antioxidative effect of CA was derived from CJ, and it was necessary to verify the antioxidative effect associated with aging using an in vivo model. Therefore, we investigated the effects of CA intervention on ROS scavenging and age-related oxidative damage during the aging process in *C. elegans*. In addition, we tried to identify the role of CA under aging-induced oxidative stress by linking it to daf-16 and skn-1.

## 2. Materials and Methods

### 2.1. Cirsium japonicum

*Cirsium japonicum* (CJ) was cultivated from the beginning of summer in an open field in Yesan (Korea). The phenolic metabolites in CJ, grown for three years, were monitored every year by HPLC. CJ was sampled annually in July, thoroughly dried in the shade, and then underwent extraction for 6 h at 60 °C with 70% ethanol using a reflux extraction method. CJ extraction was repeated three times and extracts were combined and then the solvent was removed at 50 °C using an evaporator (EYELA, Tokyo Rikakikai Co., Tokyo, Japan).

### 2.2. HPLC-DAD Analysis

Phenolic metabolites in CJ were analyzed as previously described by Jang et al. (2020) [17] using an HPLC–diode-array detector (DAD) unit (Dionex, Sunnyvale, CA, USA) equipped with a Waters Symmetry C18 column (Waters, 4.6 mm × 150 mm, 5 μm). Detailed operating conditions are provided in Appendix A. Compounds were initially identified by comparing retention times with those of reference compounds, and subsequently, by comparing UV spectra obtained using a DAD with those of reference compounds. Quantification was performed by external calibration using reference compounds. For this purpose, CA was purchased from Sigma-Aldrich (St. Louis, MO, USA) and LIN and PLIN standards were purchased from the Cheil Jedang Functional Food Research and Development Center (Seoul, Republic of Korea). Solvents used for analysis were of HPLC-grade and purchased from Millipore (Bedford, MA, USA).

### 2.3. Antioxidant Capacities

Radical scavenging ability was measured using 2,2-diphenyl-1-picrylhydrazyl (DPPH) and 2,2′-azino-bis (3-ethylbenzothiazoline-6-sulfonic acid) (ABTS), as described by Gullon et al. (2015) with modification [31]. For DPPH, briefly, the same volumes of sample and DPPH solution (0.2 mM) were mixed and placed in the dark for 30 min. The reaction mixtures were measured at 517 nm using a microplate reader (SpectraMax M2, Molecular Devices, San Jose, CA, USA).

To produce ABTS cations, a mixture containing 7.0 mM ABTS (in 20 mM sodium acetate buffer, pH 4.5) and 2.45 mM potassium persulfate was placed in the dark overnight. The ABTS solution was diluted to an absorbance value of 0.7 ± 0.01 at 734 nm. Samples (50 μL) were reacted with 950 μL of the ABTS solution for 10 min and then transferred to a 96-well plate. Absorbances were measured at 734 nm using a microplate reader.

DPPH and ABTS radical scavenging activities were calculated using the following formula:Radical scavenging activity (%) = [1 − (sample O.D./blank O.D.)] × 100

Vitamin C equivalent antioxidant capacity (VCEAC) was calculated using a VC calibration curve and are expressed as micrograms of VC per gram of sample dry weight. Additionally, IC50 representing a concentration of 50% inhibition was determined based on the % activity values of samples.

### 2.4. Protective Effects on Oxidative Stress Using C. elegans

#### 2.4.1. Nematode Culture

The *C. elegans* strains *N2* (wild-type, WT), skn-1 (*zu67*) *IV,* daf-16 (*mgDf47*), and daf-16::GFP (*zls356*) along with their food (OP50, *Escherichia coli*) were provided by the Caenorhabditis Genetics Center (CGC, University of Minnesota, Minneapolis, MN, USA). Nematodes were grown at 20 °C on nematode growth media (NGM) plates coated with OP50. All solutions required for growing nematodes, such as NGM, M9 buffer, and S-complete, were prepared according to Stiernagle T (1999) [32]. Embryos were obtained and synchronized by corroding the torsos of adult nematodes with bleach (6% hypochlorite, Clorox Company, Oakland, CA, USA). The reagents used for *C. elegans* experiments were from Thermo Fisher Scientific, Inc., (Pittsburgh, PA, USA), unless otherwise specified. The experimental methods used were adapted from Jang et al. (2020) and modified [17].

#### 2.4.2. Population Growth, Reproduction, and Pumping Rates

CA was added to 12-well plates containing synchronized L1 worms. To calculate growth rates, numbers of worms in each developmental stage were counted 2 days later. For counting the number of progenies, worms were transferred to fresh NGM plates every day during the reproduction period, and eggs were left on plates to hatch. Numbers of offspring were counted when they reached the L4 stage. To determine pumping rates, numbers of pharyngeal contractions made by 15 randomly selected nematodes were counted under a microscope for 30 s.

#### 2.4.3. Determination of ROS Levels

After culturing L1 worms for a day in an NGM plate, worms were allocated to two groups and transferred to plates. The first group was treated with CA daily for 6 days after culture for 2 days (the young group), and the second group was treated with CA for a day after culture for 7 days (the old group). ROS levels in both groups were determined on day 8. To prevent mixing generations, FUdR (120 mM) was added to the culture plates. 

Worms were then washed three times with M9 buffer, and 50 worms were transferred to 50 μL of S-complete in a 96-well plate and treated with 50 μL of fresh 100 mM DCF-DA solution in M9 buffer.

Oxidative stress resistance was evaluated using paraquat (paraQ) or juglone (JUG) assay. After culturing L1 worms for a day in NGM plates, samples were added to plates containing worms for 65 h; paraQ (4.5 mM) or JUG (80 μM) was then added to treated worms and incubated for 2 h. Worms were then washed three times with M9 buffer and treated with 100 mM DCF-DA solution as described above.

Fluorescence was measured using a fluorescence microplate reader (BioTek, Winooski, VT, USA) at excitation and emission wavelengths of 485 and 535 nm, respectively. ROS levels were expressed as percentages of those of treated worms compared to untreated worms with samples (controls).

#### 2.4.4. Antioxidant Enzyme and Catalase Activities

To determine catalase (CAT) activities, worms were prepared under JUG-induced oxidative stress. Samples were treated from L1 stage for 64 h. JUG (80 μM) was then added to treated worms and incubated for 2 h. Worms were homogenized in PBS and centrifuged at 1000× *g* for 5 min. The supernatants obtained were then assayed for CAT activities. CAT activities were measured using a CAT kit (Biomax, Guri-si, Gyeonggi-do, Republic of Korea) by measuring absorbance at 560 nm. All the results are expressed as mU per mL of protein.

#### 2.4.5. Lifespan Assay

For the lifespan assay, nematodes aged 40 h were transferred to 24-transwell plates and treated with CJ during the remainder of the lifespan study. FUdR (120 mM) was also added prevent progeny production. Lifespan assays were performed at 20 °C and media were changed every 2 days. Numbers of surviving worms were recorded every other day until all worms died. The day that treatment was started was considered to be day 0. Statistical analysis for the lifespan assay was performed using an online survival analysis application (OASIS; sbi.postech.ac.kr/oasis).

### 2.5. Statistical Analysis

The results are expressed as means ± standard deviations. The analysis was conducted using SPSS ver. 19.0 statistical analysis software (Chicago, IL, USA). Group comparisons were assessed by ANOVA (Tukey multiple range tests) using a confidence interval of 95% (*p* < 0.05). Student’s *t*-test was used to assess the significance of differences between groups. Statistical significance, as determined by the *t*-test, was accepted for *p* values < 0.05 or < 0.01, as indicated.

## 3. Results

### 3.1. Variations in Phenolic Metabolites during CJ Growth

Several phenolic compounds in CJ harvested at different times were identified by HPLC (Figure 1A–C). During the three-year harvest period, three peaks were identified, CA, LIN, and PLIN (Peak 1–3). As shown in Figure 1D, total phenolic contents and those of the three phenolics increased during the harvesting period.

### 3.2. Antioxidant Capacities of CJ

The antioxidant activities of CJ extract (25–250 µg/mL) and the three phenolics (10–100 µg/mL) were evaluated using DPPH and ABTS assays. Values were expressed as % radical scavenging activities, IC 50, and vitamin C equivalents (Table 1). In both DPPH and ABTS tests, 25 CA µg/mL showed the strongest antioxidant capacities, similar to 250 µg/mL CJ extract, followed by LIN and PLIN. Based on this result, CA, which has shown the strongest antioxidant activity among the CJ extracts, was chosen for further in vivo study.

### 3.3. Effect of CA on C. elegans Growth and Healthspan

Before investigating the efficacy of CA in *C. elegans*, we evaluated the acute toxicity of CA in *C. elegans*. The toxicity of CA was not found at concentrations below 50 μg/mL; therefore, we conducted further experiments at 25 and 50 μg/mL (Appendix A).

To determine the effect of CA on *C. elegans* growth and healthspan, we treated worms with CA (Figure 2). The growth stage distributions of nematodes treated with 25 or 50 µg/mL of CA were not significantly different from those of untreated controls (Figure 2A).

Next, the reproduction capacity of adult worms was examined (Figure 2B). Although no significant difference was observed between CA-treated worms and controls until day 3 (the young healthy adult stage), from day 4, CA-treated worms produced more progeny than untreated controls, and on day 5, old control nematodes barely produced progeny, whereas CA administration increased progeny producing capacity 5.67-fold.

The pumping rate is used to determine *C. elegans* age. As shown in Figure 2C, pumping rates decreased constantly by around 53.6% on the seventh day. Pumping rates of CA-treated (25 or 50 µg/mL) worms did not differ from controls until the seventh day, but on the fourteenth day, they were greater for treated worms than controls.

### 3.4. Lifespan Extension by CA-Induced ROS Suppression

To determine the effect of CA on *C. elegans* under normal and oxidative stress conditions, we assessed ROS generation and examined its effect on life extension. In the CA-treated (25 and 50 µg/mL) groups, ROS levels were reduced dependent on concentration (Figure 3A). Further, CA extended lifespan in a concentration-dependent manner, and maximum lifespan was extended by 6 days at 50 µg/mL (Figure 3B).

To examine the effect of CA at high ROS levels, we treated worms with 4.5 mM paraQ. The results indicated that paraQ increased ROS levels to 125%, but decreased to 110% when paraQ and CA were co-treated (Figure 3C). Additionally, we investigated whether CA could extend lifespan under oxidative stress. As shown in Figure 3D, the lifespan of nematodes exposed to ROS decreased by 50%. However, when paraQ and CA were co-treated, lifespan was extended by 4 days as compared with worms treated with paraQ only (Figure 3D).

### 3.5. CA Effects on Aging

The antioxidant effect of CA on aging was examined by treating young and old worms (Figure 4). Aging-induced worms had higher ROS levels than controls (young worms). However, CA significantly reduced ROS levels treated from young (pre-treatment) and old age (post-treatment); in particular, worms treated with CA from a young age showed less ROS accumulation than the worms treated from old age. Furthermore, CA at 50 μg/mL reduced ROS levels to that of control when treated from a young age.

### 3.6. Effect of CA on the Healthspans of daf-16 and skn-1 Transformed Worms

To understand the relationship between the action of CA and the role of transcription factors (daf-16 and skn-1) in response to oxidative damage, daf-16 and skn-1 related transformed worms were used.

As shown in Figure 5A, it was confirmed that daf-16 was translocalized from the cytoplasm to the nucleus under normal conditions in the CA-treated worm. Interestingly, however, CA did not contribute significantly to the intranuclear expression of daf-16 under oxidative stress compared to normal states. On the other hand, CA treatment on the skn-1 null worms resulted in 63% and 82% reduction in ROS expression under normal and oxidative stress conditions, respectively (Figure 5B).

Catalase activity of WT worms increased by CA under normal and oxidative conditions (Figure 5C). However, the effect of CA on catalase activity in skn-1 and daf-16 mutants was abolished (Figure 5D,E).

In addition, to accumulate mechanistic insight into CA-extended lifespan of C. elegans under oxidative stress, we evaluated the lifespans of daf-16 and skn-1 deficient mutants. As shown in Figure 5F,G, paraQ treatment reduced the lifespan of skn-1 and daf-16 mutants by 28% and 17%, respectively. As with the result of catalase activity, the lifespan-extending effect of CA in WT worms was abolished in skn-1 and daf-16 mutant worms.

## 4. Discussion

In this study, we identified the presence of CA, LIN, and PLIN in CJ and confirmed the radical scavenging effect of CA. Furthermore, in *C. elegans*, CA reduced age-induced ROS accumulation and extended lifespan. In particular, it was confirmed that these effects of CA depend on genetic pathways associated with daf-16 and skn-1.

Recently, phenolic compounds, a secondary metabolite of plants, have been in the spotlight as ingredients of dietary supplements [16,17]. two categories of research are being conducted: (1) Research on identification of maker compounds of food resources, and (2) Research on the physiological activity of each compound.

In order to develop dietary supplements containing bioactive compounds in the food industry, two categories of research are being conducted: (1) Research on the identification of maker compounds of food resources, and (2) the physiological activity of each compound [17,33].

We collected seeds from wild CJ, planted them in the field, and cultivated them for 3 years. In addition, we analyzed whether the bioactive components contained by wild CJ were also confirmed in cultivated CJ.

The biosynthesis of phenolics has been reported to be affected by various environmental factors such as UV, soil-dwelling bacteria, moisture, temperature, and biotic factors [34,35]. Three phenolic compounds, namely CA, LIN, and PLIN, were detected in CJ grown for three years, consistent with substances contained in wild CJ (Appendix A). In addition, during these three years, total phenolic contents and individual contents of CA, LIN, and PLIN in CJ extracts gradually increased. Gao et al. (2020) harvested *Stauntonia hexaphylla* leaves at different years and found that levels of some bioactive compounds and the antioxidant effect were increased from the first to third years [35]. Regarding that, it has been reported that plants develop their antioxidant defense system to resist environmental stresses and produce phenolics as secondary metabolites, affecting their defense abilities against oxidation [35,36,37]. Therefore, we hypothesized that the total phenol content might be increased by the plant’s intrinsic defense system during cultivation and that three major phenolics of CJ might influence biological activity.

In the present study, CA had greater DPPH and ABTS radical scavenging activities than LIN or PLIN, which suggests CA plays a major antioxidative role in CJ. During the three harvesting years, CJ contained the highest amount of LIN, which was not correlated to the radical scavenging activities of the compounds. In a study by Djeridane et al. (2006), antioxidant activity was not always explained by content but rather by chemical scavenging activities [38]. In addition, Balasundram et al. (2006) reported the antioxidant activities of phenolics are due to their radical scavenging effects, and that these depend on structural differences [39].

Aging is characterized by declines in tissue functions and progressive deterioration in muscle function and physical health [1]. Several biomarkers used to examine antiaging capacity reflect physiological and functional degenerations of organisms [40,41]. *C. elegans* is considered a useful in vivo model for genetic aging studies due to its short life cycle [11]. In the present study, we used it to investigate beneficial effects attributable to the antioxidative effects of CA. The reproductive ability of hermaphrodites such as *C. elegans* is a marker of physiological status and declines with age. Normally progeny numbers peak on day 2 of adulthood and rapidly decline from day 3 [40,42]. Pharyngeal pumping is a movement of pharyngeal muscles when *C. elegans* consumes food and this movement progressively declines with age [43,44]. In our study, though CA treatment did not affect population growth, it did attenuate age-related reductions in progeny numbers and pumping rate. These findings indicate that CA can attenuate the age-related physical decline in *C. elegans* and that this is related to its antioxidative effects.

ROS are byproducts of cellular respiration and are normally scavenged by the antioxidative defense system [3]. However, excessive ROS accumulation causes oxidative stress, impairs stress resistance mechanisms, and destroys DNA, protein homeostasis, and cellular signaling [45]. During aging, mitochondria homeostasis is altered and ROS scavenging efficacy decreases, which leads to ROS accumulation, resulting in serious and progressive aging-related diseases [46].

In a study by Tungmunnithum et al. (2020), ROS levels were 3.5 higher in aged yeast cells (5 days of cultivation) than in young yeast cells (2 days of cultivation) and CA treatment effectively and dose-dependently reduced ROS levels [3]. These results were also confirmed in our *C. elegans* study. We found that the progeny reproduction capacity was almost eliminated on day 5 of nematode culture and the pumping rate was reduced by 50% on day 7. Therefore, worms cultured for 7 days were defined as old worms, and the effect of CA treatment on ROS levels was evaluated at different times. CA treatment inhibited ROS accumulation in both pre- and post-treatment worms, especially in the pre-treatment group with CA at 50 μg/mL, which was reduced to the control (young worms) level. These results indicate that aging could be a factor to elevate ROS levels, and antioxidant treatment from a young age better protects against oxidative stress [3]. In this study, therefore, we focused on the gradual oxidation during the aging process and the effect of the dietary antioxidant CA intervention.

As a result, CA lowered ROS accumulation under normal and elevated oxidative stress. In addition, CA effectively extended lifespan and delayed aging under oxidative stress. Our results suggest that ROS accumulation is affected only by the aging process, and CA intervention has shown a preventive effect on ROS damage due to aging.

skn-1 (A Nrf2 homolog) is a transcription factor for oxidative stress response and lifespan extension [47]. skn-1-related genes induce resistance to oxidative stress and aging by scavenging radicals [48,49]. daf-16 is a FOXO-family transcriptional factor in the insulin/IGF-1-like (IIS) signaling pathway, which is a key regulator of genes involved in the development, stress resistance, thermal tolerance, metabolism, and lifespan [50]. skn-1 and daf-16 have been commonly reported as prerequisites for activating antioxidant genes such as Mn-SOD (*sod-3*) and catalase (*ctl-1* and *ctl-2*) [51]. Therefore, both transcription factors belong to the antioxidant system and respond to oxidative stress. In our results, the effect of CA on catalase activity in WT was eliminated in the skn-1and daf-16 mutants, which means that both transcription factors regulate catalase activity.

Meanwhile, Tullet et al. (2008) demonstrated that skn-1 works on antioxidant defense and longevity independently of daf-16 [52]. Although there have been a few reports that the effect of CA depends on daf-16 and skn-1, we needed to investigate how these two transcription factors (daf-16 and skn-1) work in the gradual oxidation reaction of aging.

In our study, lifespan extension through CA treatment was obviously abrogated in the absence of skn-1 gene, but there was partial effect in the absence of the daf-16 gene. To understand the effect of CA in the natural course of aging, we evaluated the CA effect on the nuclear localization of daf-16 related to antiaging and antioxidant mechanisms and found an interesting role of CA in the workings of daf-16. In our results, the body ROS contents in the aged nematodes were larger than in young nematodes. Additionally, the role of CA on daf-16 migration into the nucleus was greater under oxidative stress than under basal condition. Assuming aging as a kind of oxidative stress, this means that CA linked with daf-16 may actively work on the antioxidative system of healthy young condition before age. Several studies have noted that under moderate oxidative stress conditions, the transition of daf-16/FOXO from the cytoplasm to the nucleus is stimulated, activating genes that play a role in the stress response. [13,53]. We further evaluated the effect of CA on ROS generation in the skn-1 knockout mutant to investigate the role of skn-1in the aging phase. Interestingly, daf-16 was less responsive to strong oxidative stress than to moderate oxidative stress, while skn-1 responded even under extreme oxidative stress. These results suggest that daf-16 and skn-1 have their own roles independently under oxidative stress caused by natural aging. 

Future studies require a prerequisite for subdividing the aging stage to determine when daf-16 and skn-1 are at work. In addition, research needs to be conducted to identify the molecular mechanism responsible for CA in the downstream pathways of daf-16 and skn-1 at each stage of aging.

## 5. Conclusions

In the present study, we investigated the potent antioxidant effect of CA contained in CJ and its effect on age-induced oxidative stress. CA showed the most radical scavenging effect among the phenolics in CJ extract.

In the *C. elegans* model, CA treatment significantly increased the numbers of progeny and the pumping rate. Moreover, CA extended lifespan and inhibited ROS accumulation under normal and oxidative stress conditions. The inhibition effect of CA on ROS was also shown against the produced ROS during the actual aging process, and the inhibitory effect was stronger when CA was treated from an early age.

Further studies were conducted to understand the role of CA against actual aging and oxidative stress in association with daf-16 and skn-1. CA strongly induced the expression of daf-16 in the nucleus under normal condition compared to oxidative stress. On the other hand, it was found that the reactivity of CA to ROS was greatly increased in skn-1 mutants under both normal and oxidation conditions. In addition, the increased effect of CA on catalase activity and lifespan in WT was abrogated in daf-16 and skn-1 mutants. This suggests that the expression of both transcription factors modulates the antioxidant activity of CA in vivo and positively affects longevity.

Taken together, daf-16 reacted to mild oxidative stress and skn-1 reacted to overall oxidative stress for the activation of the antioxidant system. In other words, if aging is divided into early (mild oxidative stress) and late (accumulated oxidative stress) stages depending on the intensity of oxidative stress, daf-16 and skn-1 might play their independent roles in the early and late stages of aging.

## Figures and Tables

**Figure 1 metabolites-13-00224-f001:**
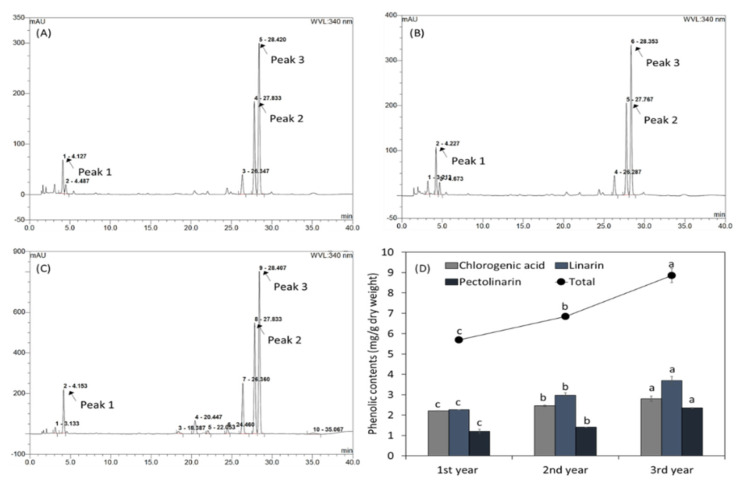
High—performance chromatography profiles of the phenolics in CJ cultivated and harvested in a field for three years: (**A**) first year, (**B**) second year, and (**C**) third year; peak 1: CA, peak 2: LIN, and peak 3: PLIN. (**D**) Total phenolic contents of CJ cultivated in a field for three years. Values are means ± standard deviations of three separate experimental results (n = 3). The letters on the same color bars (a to c) presented represent statistically significant differences (*p* < 0.05).

**Figure 2 metabolites-13-00224-f002:**
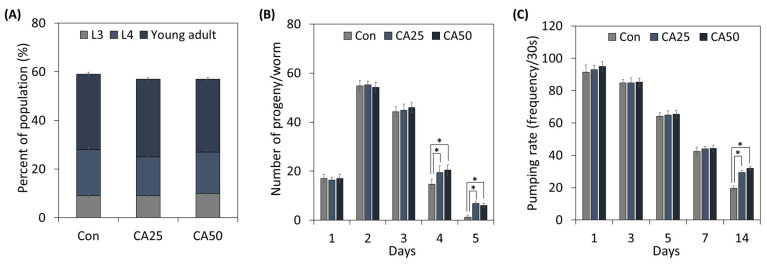
Effects of CA on *C. elegans* growth and aging factors: (**A**) population growth, (**B**) progeny production, and (**C**) pumping rate. From the L1 stage, nematodes were treated with 25 or 50 μg/mL CA. Values represent means ± standard errors (n = 3 plates of 100 worms/plate for growth rate; n = 3 plates and 5 worms/plate for progeny production; n = 3 plates and 15 worms/plate for pumping rate). * means a significant difference between the values of two bars (*p* < 0.05).

**Figure 3 metabolites-13-00224-f003:**
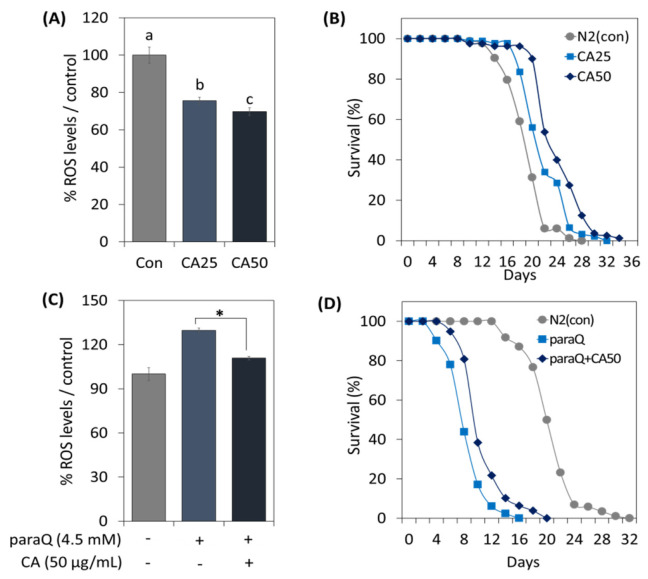
Effects of CA on oxidative stress and lifespan extension in *C. elegans*. Effects of CA (25 or 50 μg/mL) on ROS generation under (**A**) normal and (**C**) oxidative stress conditions. Survival curve of *C. elegans* treated with CA (25 or 50 μg/mL) under (**B**) normal and (**D**) oxidative stress conditions. Bars represent the means ± SDs of three separate experiments. Letters above bars indicate significant differences (*p* < 0.05). For the survival study: (**B**,**D**), worms were treated with CA from the young adult stage (day 0) and survivals were recorded every other day until all died. Note: n = 3 plates and 78–91 worms/group. * means a significant difference between the values of two bars (*p* < 0.05).

**Figure 4 metabolites-13-00224-f004:**
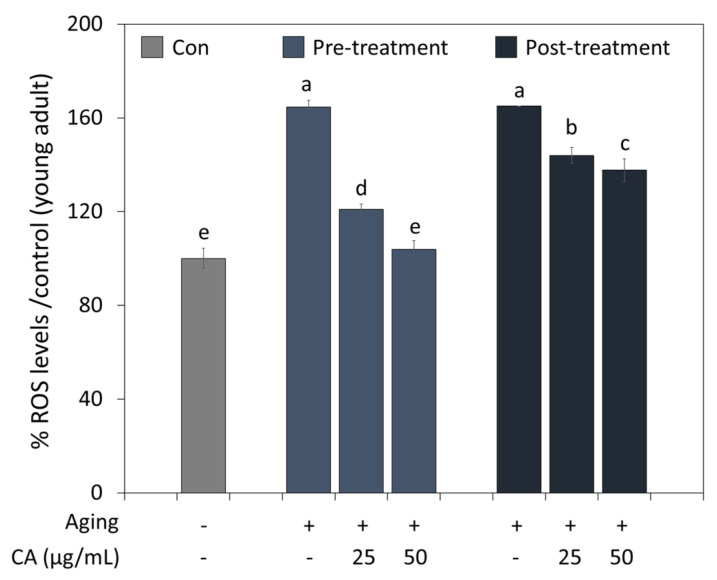
Pre- or post-treatment effects of CA (25 or 50 μg/mL) on ROS generation during *C. elegans* aging. Worms in the pre- and post-treatment groups were treated with CA starting from the young adult stage (day 2) or the aged adult stage (day 7), respectively. Letters above bars indicate significant differences (*p* < 0.05).

**Figure 5 metabolites-13-00224-f005:**
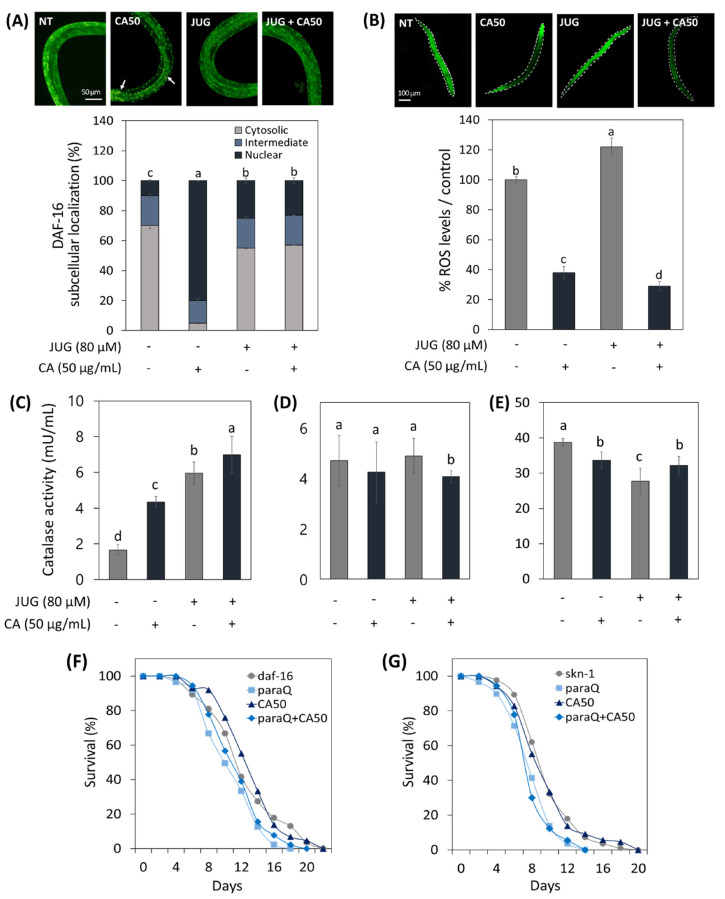
Effects of CA (50 μg/mL) on ROS-related aging via skn-1/Nrf2 and daf-16/FOXO. (**A**) CA effect on daf-16 nuclear localization in daf-16::GFP worms. (**B**) CA effect on ROS generation in skn-1 knockout mutant. CA Effect on catalase activity in (**C**) N2, (**D**) daf-16 mutants, and (**E**) skn-1 mutants under normal and oxidative stress condition. Survival curve of (**F**) daf-16 and (**G**) skn-1 mutants treated with or without CA (50 μg/mL) under normal and oxidative stress. Oxidative stress was induced using 4.5 mM paraquat (paraQ) or 80 μM juglone (JUG). NT: not treated. Bars represent the means ± SDs of three separate experiments (n = 3). Letters above bars indicate significant differences as determined by Duncan’s multiple range test (*p* < 0.05). For the survival study, *C. elegans* was treated with CA starting from the young adult stage (day 0) and the survivals were recorded every other day until all worms died. Note: n = 3 plates and 78–91 worms/group.

**Table 1 metabolites-13-00224-t001:** Antioxidant properties of CJ extract and its individual phenolics.

Samples	DPPH						
	Radical Scavenging Activity (%) of Samples (µg/mL)					IC50	VCEAC
	10	25	50	100	250		
CJ ext	NT	3.2 ± 0.2 ^Cd^	8.5 ± 2.5 ^Dc^	28.2 ± 1.2 ^Db^	70.0 ± 0.5 ^a^	181.3 ± 0.5 ^A^	12.0 ± 0.5 ^A^
CA	32.0 ± 0.3 ^Ad^	52.1 ± 0.3 ^Ac^	80.1 ± 0.5 ^Ab^	85.8 ± 0.4 ^Aa^	NT	24.0 ± 0.5 ^D^	11.8 ± 0.8 ^A^
LIN	10.1 ± 0.3 ^Bd^	24.3 ± 0.3 ^Bc^	52.3 ± 0.3 ^Bb^	67.1 ± 0.3 ^Ba^	NT	49.8 ± 0.3 ^C^	4.1 ± 1.2 ^B^
PLIN	9.9 ± 0.0 ^Bd^	23.5 ± 0.4 ^Bc^	48.2 ± 0.2 ^Cb^	58.9 ± 0.1 ^Ca^	NT	55.7 ± 0.2 ^B^	3.6 ± 1.2 ^C^
	**ABTS**						
	**Radical scavenging activity (%)** **of** **samples** **(µg/mL)**					**IC50**	**VCEAC**
CJ ext	NT	2.2 ± 0.2 ^Da^	6.3 ± 0.5 ^Da^	25.82 ± 1.5 ^Da^	54.0 ± 0.2	93.3 ± 0.2 ^A^	40.1 ± 0.5 ^A^
CA	12.1 ± 0.5 ^Ac^	53.8 ± 0.8 ^Ab^	72.2 ± 0.6 ^Aa^	73.6 ± 2.5 ^Aa^	NT	27.7 ± 0.5 ^B^	40.3 ± 0.5 ^A^
LIN	5.8 ± 0.2 ^Cd^	8.6 ± 0.8 ^Cc^	19.2 ± 0.8 ^Cb^	28.9 ± 0.3 ^Ca^	NT	ND	27.2 ± 0.8 ^B^
PLIN	7.2 ± 0.1 ^Bd^	12.2 ± 1.8 ^Bc^	23.8 ± 1.5 ^Bb^	32.1 ± 0.7 ^Ba^	NT	ND	26.0 ± 2.0 ^BC^

Values are means ± SDs of three separate experimental results (n = 3). A small letter means statistically significant differences by concentration (*p* < 0.05). A large letter means statistically significant differences between samples. IC50: 50% inhibitory concentration. Vitamin C equivalent antioxidant capacity (VCEAC) was calculated using a VC calibration curve and is expressed as micrograms of VC per gram of sample dry weight. CJ ext: *C. japonicum* extract, CA: chlorogenic acid, LIN: linarin, and PLIN: pectolinarin. NT: not tested; ND: not detected.

## Data Availability

Data available on request due to restrictions eg privacy or ethical.

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
