# Peer review of "Chlorogenic Acid of Cirsium japonicum Resists Oxidative Stress Caused by Aging and Prolongs Healthspan via SKN-1/Nrf2 and DAF-16/FOXO in Caenorhabditis elegans"

_metabolites, 2023, doi:10.3390/metabo13020224_

Round 1

Reviewer 1 Report

This study aims to evaluate the healthy longevity effect of Cirsium japonicum by in vitro and in vivo methods. This topic is meaningful but there are already some similar studies. The novelty of this paper was not explained well. Chlorogenic acid has been widely reported to exhibit antioxidant activity by activating Nrf2 and prolong lifespan in the C. elegans model. Moreover, some conclusions were not supported by enough experimental data to elucidate the mechanism clearly, and the result analysis and discussion were insufficient. In addition, the paper was not organized and presented well. The detailed comments are as follows.

1.     Results 3.1, Main compounds of Cirsium japonicum has already been reported such as chlorogenic acid, linarin and pectolinarin. What’s the meaning of measure phenolic metabolites during CJ growth?

2.     Figure 4, where are A and B? There is only one figure.

3.     Figure 6, what’s the concentration of juglone, 70 or 80 μM? Why chose paraquat for survival assay and juglone for enzymatic activity and nuclear localization assays? Is there any data about SKN-1 nuclear localization?

Author Response

This study aims to evaluate the healthy longevity effect of Cirsium japonicum by in vitro and in vivo methods. This topic is meaningful but there are already some similar studies. The novelty of this paper was not explained well. Chlorogenic acid has been widely reported to exhibit antioxidant activity by activating Nrf2 and prolong lifespan in the C. elegans model. Moreover, some conclusions were not supported by enough experimental data to elucidate the mechanism clearly, and the result analysis and discussion were insufficient. In addition, the paper was not organized and presented well. The detailed comments are as follows.

Results 3.1, Main compounds of Cirsium japonicum has already been reported such as chlorogenic acid, linarin and pectolinarin. What’s the meaning of measure phenolic metabolites during CJ growth?

We have tried to develop health-functional food using thistle through many studies for a long time. In our earlier study, we identified linnarin, pectorinarin, and chlorogenic acid in collected wild thistles. After then, we collect their seeds, planted thistles in the open field, and farmed them for three years. When wild thistle seeds grew into plants, it was necessary to ensure whether wild thistle-specific active substances were maintained and to identify changes in their contents during the growth period. We observed that the content of functional substances also increased as the cultivation period increased during the cultivation period of three years. We added this research background to the discussion.

Figure 4, where are A and B? There is only one figure.

We corrected it.

Figure 6, what’s the concentration of juglone, 70 or 80 μM? Why chose paraquat for survival assay and juglone for enzymatic activity and nuclear localization assays? Is there any data about SKN-1 nuclear localization?

We treated 80 μM Juglone to cause oxidative damage. Figure 6 has been modified.

Both paraquat and juglone induce oxidative damage. We initially used paraquat to induce oxidative damage. However, in the middle of the study, paraquat was designated as a prohibited chemical in Korea and could not be used for subsequent experiments. Therefore, after then, we used Juglone to induce oxidative damage.

We did not observe the intranuclear expression of SKN-1. However, the body ROS content was analyzed using the SKN-1 knockout mutant, and the result was added in Figure 6.  Also, the interpretation was added to the discussion.

All modified sentences are marked in red.

Reviewer 2 Report

In this study, Cho and colleagues aimed to evaluate the effect of chlorogenic acid isolated from Cirsium japonicum on oxidative stress induced by aging as well as the life span of C. elegans. After a careful review, several drawbacks, in particular the lack of novelty, have been encountered.

The study lacks novelty because the antioxidant and anti-aging effect of chlorogenic acid in C. elegans has been previously studied even on a more detailed level. For example, the study of Zheng et al (2017 -  https://doi.org/10.1093/gerona/glw105). I do realize that one of the aims of the study was to check the phenolic content and CA in Cirsium japonicum on over time but this is still of no novelty.

The Abstract should be rewritten following the appropriate style: Background, aim, methods, results, and conclusion.

The introduction is very poorly written Please provide an appropriate background showing the redox imbalance in aging and its negative impact on the body.

The paragraph on chlorogenic acid is not enough for the readership to know what is chlorogenic acid, its sources, benefits, pharmacological effects …… etc.

There should be at least one paragraph on SKN-1/Nrf2 and its role in counteracting oxidative stress particularly in the context of aging.

The Introduction section should be re-written to improve readability.

In the DPPH and ABTS assays, the concentration-dependent inhibitory activity should be conducted in order to determine the IC50.

In the same assays, it is not clear how much the authors used. They just mentioned sample volume.

In Figure 2 (DPPH and ABTS assays), the Y-axis title it inappropriate. This axis should represent the inhibition %.

The quality of the figures should be improved.

The authors treated the worms with 25 or 50 µg/mL of CA. (a) The use of these specific concentrations should be justified. (b) is there a relation between these concentrations and the concentrations used in the in vitro radical-scavenging assays?

What are the changes in antioxidant defenses encountered in the mutant worms?

The discussion should be more focused and more detailed and not just repeating the results and mention the studies in agreement.

Author Response

[Reviewer 2]

In this study, Cho and colleagues aimed to evaluate the effect of chlorogenic acid isolated from Cirsium japonicum on oxidative stress induced by aging as well as the life span of C. elegans. After a careful review, several drawbacks, in particular the lack of novelty, have been encountered.

The study lacks novelty because the antioxidant and anti-aging effect of chlorogenic acid in C. elegans has been previously studied even on a more detailed level. For example, the study of Zheng et al (2017 -  https://doi.org/10.1093/gerona/glw105). I do realize that one of the aims of the study was to check the phenolic content and CA in Cirsium japonicum on over time but this is still of no novelty.

As you mentioned, I wanted to talk about the increase in CA content as the cultivation period increased. However, the most important thing to point out was the role of CA on the production body ROS not only under oxidative stress conditions but also during the actual aging process. Previous studies have also confirmed the role of CA in longevity under various conditions. Research on aging and lifespan is considered similar, but it is necessary to understand aging and lifespan separately because lifespan varies depending on the aging process. Our results showed that ingesting CA induces extension of lifespan by reducing the body ROS content during aging.

The Abstract should be rewritten following the appropriate style: Background, aim, methods, results, and conclusion.

We rewrote the abstract as your advice.

The introduction is very poorly written Please provide an appropriate background showing the redox imbalance in aging and its negative impact on the body.

In the introduction, the relationship between excessive accumulation of ROS in the body and aging was added.

The paragraph on chlorogenic acid is not enough for the readership to know what is chlorogenic acid, its sources, benefits, pharmacological effects …… etc.

We added general and scientific information about chlorogenic acid in the introduction.

There should be at least one paragraph on SKN-1/Nrf2 and its role in counteracting oxidative stress particularly in the context of aging.

We added the role of SKN-1/nrf-2 under oxidative stress related to aging.

The Introduction section should be re-written to improve readability.

We rewrote the introduction as you advised.

In the DPPH and ABTS assays, the concentration-dependent inhibitory activity should be conducted in order to determine the IC50. In the same assays, it is not clear how much the authors used. They just mentioned sample volume.

Figure 2 was modified as a table, and the table was filled with % activity and IC50. The concentration of the sample used was indicated in the added table.

In Figure 2 (DPPH and ABTS assays), the Y-axis title it inappropriate. This axis should represent the inhibition %.

Figure 2 was modified as a table, and the table was filled with % activity and IC50.

The quality of the figures should be improved.

The figures were made as high resolution required by your journal. However, the figures may not be clearly visible because they were reduced according to the manuscript format. If you point out the figures that need to be modified in size and layout, I will modify them.

The authors treated the worms with 25 or 50 µg/mL of CA. (a) The use of these specific concentrations should be justified. (b) is there a relation between these concentrations and the concentrations used in the in vitro radical-scavenging assays?

We evaluated acute toxicity using nematodes, and as a result, a non-toxic concentration was selected and used in the experiment. The result was added to the supplementary materials.

What are the changes in antioxidant defenses encountered in the mutant worms? The discussion should be more focused and more detailed and not just repeating the results and mention the studies in agreement.

In the discussion section, we wrote a more in-depth focus on the oxidation defense system in mutant worms.

Reviewer 3 Report

Remarks to the Author:

Myogyeong Cho and colleagues investigated investigated the potent antioxidant of chlorogenic acid and its effect on age-induced oxidative stress and found that chlorogenic acid of Cirsium japonicum resists oxidative stress caused by aging and prolongs healthspan via SKN-1/Nrf2 in Caenorhabditis elegans. It is an interesting work research, but there are many pitfalls and caveats from the presented results.

Overall, I think the manuscript should be reconsidered after major revision.

Major points:

1. Abstract needs to be rewritten. The abstract usually contains the following contents: background or research purpose of the project, research content and research methods, main findings and conclusions, impact of research or future direction.

2. Introduction needs to be written more comprehensively. For example, why do you use C. elegans. as an animal model? It is necessary to add relevant descriptions in introduction.

3. Describe the magnification or add the scale bar in Figure 6.

4. In Figure 6F, please describe what NT is and what strain of nematode is used.

5. How to choose the exposure mode (solid exposure or liquid exposure), exposure concentration and exposure time of CA? Why choose CA of this concentration (25 or 50 μg/mL)? Additional experiments are needed or the reasons for such selection in the text should be clarified.

6. Is there any statistics on fluorescence intensity of Figure 6F? It is necessary to add the fluorescence statistical chart of Figure 6F.

7. line 253: “As with the result of the catalase activities, it was confirmed that the lifespan extension effect of CA was abolished in skn-1 and daf-16 mutant worms. In the present study, the longevity effect due to CA was partially affected in daf-16 mutants, which was abolished distinctly in skn-1 mutants. The CA effect on daf-16 expression in the nucleus was stronger under normal condition than under oxidative stress (Figure 6F).”. The lifespan extension effect of CA was abolished in skn-1 and daf-16 mutant worms was confirmed by catalase activities rather than survival curve, what is the relationship between catalase activities and lifespan? can you explain the reasons?

Author Response

[Reviewer 3]

Myogyeong Cho and colleagues investigated investigated the potent antioxidant of chlorogenic acid and its effect on age-induced oxidative stress and found that chlorogenic acid of Cirsium japonicum resists oxidative stress caused by aging and prolongs healthspan via SKN-1/Nrf2 in Caenorhabditis elegans. It is an interesting work research, but there are many pitfalls and caveats from the presented results.

Overall, I think the manuscript should be reconsidered after major revision.

Major points:

  1. Abstract needs to be rewritten. The abstract usually contains the following contents: background or research purpose of the project, research content and research methods, main findings and conclusions, impact of research or future direction.

We rewrote the abstract as your advice.

  1. Introduction needs to be written more comprehensively. For example, why do you use C. elegans. as an animal model? It is necessary to add relevant descriptions in introduction.

We added comprehensive content related to this study to the introduction.

  1. Describe the magnification or add the scale bar in Figure 6.

We added the scale bar to the microscope image.

  1. In Figure 6F, please describe what NT is and what strain of nematode is used.

NT means 'not treated' and we added in the figure caption. And the strains of nematode used were listed in the ‘materials and methods’ section and information was added to the figure caption about the strains of used nematode for Figure 6.

  1. How to choose the exposure mode (solid exposure or liquid exposure), exposure concentration and exposure time of CA? Why choose CA of this concentration (25 or 50 μg/mL)? Additional experiments are needed or the reasons for such selection in the text should be clarified.

We evaluated acute toxicity using nematodes, and as a result, a non-toxic concentration was selected and used in the experiment. The result was added to the supplementary data.

  1. Is there any statistics on fluorescence intensity of Figure 6F? It is necessary to add the fluorescence statistical chart of Figure 6F.

The image of the fluorescence intensity was quantified and presented as a graph, and statistical differences were displayed on the graph.

  1. line 253: “As with the result of the catalase activities, it was confirmed that the lifespan extension effect of CA was abolished in skn-1 and daf-16 mutant worms. In the present study, the longevity effect due to CA was partially affected in daf-16 mutants, which was abolished distinctly in skn-1 mutants. The CA effect on daf-16 expression in the nucleus was stronger under normal condition than under oxidative stress (Figure 6F).”. The lifespan extension effect of CA was abolished in skn-1 and daf-16 mutant worms was confirmed by catalase activities rather than survival curve, what is the relationship between catalase activities and lifespan? can you explain the reasons?

The relationship between daf-16 and catalase in extending lifespan was added to the discussion section.

Thanks for your specific comments. In the revised manuscript, all modified sentences are marked in red.

Round 2

Reviewer 1 Report

The revised manuscript has been improved a lot. The authors have answered my concerns and revised accordingly. No more comments.

Reviewer 2 Report

The authors introduced the necessary changes 

Reviewer 3 Report

The author and colleagues have answered and revised all the questions raised, and the writing logic have been revised carefully. Therefore, I support the publication of this study.